# Zoonotic Risks of Sleeping with Pets

**DOI:** 10.3390/pathogens11101149

**Published:** 2022-10-05

**Authors:** Lucie A. Zanen, Johannes G. Kusters, Paul A. M. Overgaauw

**Affiliations:** 1Department Population Health Sciences, Division of Veterinary Public Health, Institute for Risk Assessment Sciences, Faculty of Veterinary Medicine, Utrecht University, P.O. Box 80178, 3508 TD Utrecht, The Netherlands; 2Department of Medical Microbiology, University Medical Center Utrecht, P.O. Box 85500, 3508 GA Utrecht, The Netherlands

**Keywords:** bedroom, dog, cat, pets, fleas, Enterobacteriaceae

## Abstract

Background: Pets are increasingly becoming part of the family and interactions between pets and their owners is changing. This results in extended and more intimate contact between owners and their pets, which give rise to zoonotic risks. Objective: To establish the presence of potential zoonotic pathogens in pets that sleep with their owner. Methods: As a pilot study, a group of 28 healthy dogs and 22 healthy cats were monitored for the presence of the zoonotic parasites Cheyletiella, *Ctenocephalides* spp. and *Toxocara* spp., the dermatophyte *Microsporum canis*, and the bacteria *Clostridium difficile*, *Salmonella* spp., *Campylobacter jejuni* and Enterobacteriaceae. This was investigated by taking samples from the fur, the footpads and the animal bed. The owners filled in a questionnaire. Results: In total, 29 of the 50 pets (58%) slept on the bed, of which 15 pets (30%) slept in the bed (under the blankets). A total of 19/22 dogs (86%) and 7/22 cats (32%) tested positive for Enterobacteriaceae on the fur or footpads. Fleas were found in 5/22 of the cats’ (23%) and 2/28 of the dogs’ (7%) favourite sleeping spots. High levels of aerobic colonies were found, up to 216 colony forming units/cm^2^. Other pathogens were not found in this study. Conclusions: The results of this preliminary study confirm literature reports that pets may constitute a potential risk in the transmission of zoonotic pathogens to their owner, especially during direct contact when sleeping in the same bed. Owners should therefore be informed about these risks and educated to interact with their pets in a more responsible way.

## 1. Introduction

In many industrialised countries, pet ownership is still increasing. In 2021, it was estimated that 90 million European households (45% of all households) owned at least one pet animal; 25% of households owned dogs and 25% owned cats. There were 93 million pet dogs and 114 million pet cats in Europe, which is a 26% increase in dogs within 11 years (74 million in 2010) and a 34% increase in cats (85 million in 2010) [1]. 

Dogs and cats are no longer considered workers, to keep guard or catch mice, and only allowed to sleep in the barn, but are increasingly considered members of the family [2]. They now live in our homes, sit on our laps, lick our faces and are almost treated like humans [3]. People believe that animals have awareness, thoughts and feelings and this behaviour is called anthropomorphism, personification, or humanisation.

An increasing number of owners take this “humanisation” or “anthropomorphism” as far as allowing their pets to sleep in their bed with them. More excessive forms of anthropomorphism became clear in a study for the presence of zoonotic parasites in healthy dogs and cats in the Netherlands. Half of the owners allow pets to lick their faces. Sixty percent of pets visit the bedroom; 45% of dogs and 60% of cats are allowed on the bed and 18% of dogs and 30% of cats sleep with their owner in bed. Six percent of pets always sleep in the bedroom [4]. 

Pets can be beneficial to their owners by giving psychological support and relieving stress [5,6], however, they do not pay attention to where they walk outside and do not wipe their feet after arriving home. Dogs and cats regularly lick their anus/genitals and thereafter the fur, while dogs often eat poop (coprophagia) and like to roll in carcasses [7]. Pets may be infected therefore with a variety of zoonotic pathogens to which the owner may be particularly exposed if sharing a bed with them [8]. Transmission of these pathogens could lead to serious illness, including the plague [9,10]. 

It is not only sleeping with pets that exposes owners to any pathogens that they might be carrying. Exposure can occur simply by allowing them in the home and petting them. However, sharing a bed with a pet means a higher exposure rate and therefore an increased risk of contracting zoonotic infections. 

Aside from these zoonotic risks, there are risks of dog and cat bites, scratches, allergic reactions, and the transmission of vector-borne diseases. Although pets do not transmit arthropod-borne diseases to people (such as ehrlichiosis, anaplasmosis and Lyme borreliosis), they do bring zoonotic disease vectors, such as ticks and fleas, in close proximity to people, e.g., when they sleep next to their owners [11]. In general, more than 90% of the fleas of veterinary importance on both cats and dogs are cat fleas, *Ctenocephalides felis*. Cat fleas transmit Bartonella henselae, the causative agent of cat-scratch disease, from cat to cat and from the cat to the human via infected flea feces. They also transmit B. clarridgeiae that can cause similar symptoms as B. henselae in humans. Moreover, significant increases of dermatophytosis or ringworm, a common zoonotic fungal skin infection in mainly children, caused by *Microsporum* spp., *Trichophyton* spp. or *Arthroderma* spp., has been often reported where the presence of pets is always mentioned. Contact with infected pets in bed may increase the infection risk [7,11].

Risk analysis is the result of multiplying the outcome of hazard characterisation, the impact of getting infected and exposure assessment. Hazard characterisation includes the prevalence of animals as a reservoir, their virulence to man, transmission routes and survival of the agent in the environment. The impact includes the seriousness of a disease, the chance of complications and the economic consequences that may be expected. It has been calculated in disability-adjusted life years (DALYs). The exposure assessment concludes who is exposed, for how long, how often and how much of the potential pathogen is needed to become a health risk [12]. 

We therefore conducted a pilot study to determine the potential exposure to zoonotic pathogens when owners share their bed with pets. 

## 2. Methods

### 2.1. Selection of The Owners

Owners of clinically healthy dogs and cats were randomly recruited among students, employees, families and friends in the region of Utrecht, the Netherlands, to volunteer in this study. The participants owned animals in different living situations, varying from dorm rooms to family houses, and ranged from one student to full families. Ethics approval was not required. The study was conducted during the months February and March.

### 2.2. Questionnaires

All owners were questioned about their own health and the health of their pets, the sleeping habits of their pet, the diet fed and parasite control (Table 1).

### 2.3. Sampling of the Animals

The presence of the ectoparasites Cheyletiella (fur mite) and *Ctenocephalides* spp. (fleas), the endoparasite *Toxocara* spp. (dog and cat roundworm), the dermatophyte *Microsporum canis*, and the bacteria Clostridium difficile, *Salmonella* spp., Campylobacter jejuni, and the group of Enterobacteriaceae was investigated. These pathogens are easily detectable and may act as sentinel pathogens. All the pathogens are proven to be zoonotic and are present in European domestic dogs and cats [13,14]. All animals were sampled at home.

#### 2.3.1. Fur

Sampling of the fur was performed with a latex glove moistened with 1 mL 0.1% polysorbate 80, peptone-saline solution. The animal’s head and back were stroked, followed by both flanks (stroking the flanks back and forth, attempting to reaching the deeper layers of the hair to the skin). The glove was taken off inside out and filled with 10 mL of the 0.1% polysorbate 80, peptone-saline solution and tied closed. The back of the animal was combed with a chlorine disinfected flea comb, until 15 or more hairs were collected.

#### 2.3.2. Foot Pads

An imprint of the animal’s front foot was taken by gently pressing the sole of the foot on a Violet Red Bile Glucose (VRBG) agar plate for 1 s.

#### 2.3.3. Animal Beds

The regular sleeping place of the animal was sampled at the home of the owner by shaking or beating the cushions or blankets above a plastic-coated piece of paper. The obtained material was collected into a petri dish.

### 2.4. Analysis of The Collected Materials

Samples were stored for maximum of 24 h in the refrigerator at 4 °C and then processed in the lab.

#### 2.4.1. Fur Material

Bacterial investigation: The latex glove was thoroughly shaken and kneaded, after which 1 mL was applied on an Enterobacteriaceae petrifilm (3M™ Petrifilm™ EB) and 1 mL on three aerobic count petrifilms (3M™ Petrifilm™ AC) in a dilution of 1:1, 1:10 and 1:100, respectively. The EB Petrifilm was incubated for 24 h at 37 °C, the AC Petrifilms were incubated for three days at 30 °C. After this period, the number of colonies were counted (on the EB Petrifilm only Enterobacteriaceae colonies were counted). After incubation, the 1:1 diluted AC Petrifilm was swabbed. The swab was stored in Cobas buffer until it was investigated using polymerase chain reaction (PCR) for the presence of *Clostridium difficile*, *Salmonella* spp. and *Campylobacter jejuni* (Gastro Bacterial Lightmix kit by TIB Molbiol, Clostridium RIDAGENE Clostridium difficile & Toxain A/B kit, Salmonella ltr target, Campylobacter 16s. Protocol and analysis according to routine clinical diagnostics in Lightcycler 480II PCR machine). Besides this, eight randomly picked glove samples were plated on blood agar TSA-S plates and Campylobacter blood free selective medium and CCDA selective supplement SR0155E plates, incubated for four days, respectively, at 37 °C for the blood agar and 42 °C, micro aerophilic, for the Campylobacter plates. The colonies growing on the blood plates were tested using a MALDI Biotyper^®^.

Fungal investigation: The hairs were deposited with chlorine-disinfected tweezers on an agar medium for the culture of pathogenic fungi (Dermatophytest™, Virbac, Barneveld, The Netherlands). The Dermatophytest^TM^ was used according to the manufacturer’s instructions and was checked every other day for change in colour, indicating dermatophyte growth.

#### 2.4.2. Footprints

The VRBG agar plate with the footprint was incubated for 24 h at 37 °C under aerobic conditions, after which the number of Enterobacteriaceae colonies was counted.

#### 2.4.3. Animal Bed Material

Petri dishes with material from the animal’s regular sleeping place were first examined using a stereo microscope with 10–50× magnification for flea stages and Cheyletiella mites (and eggs). Then, the material was investigated with a flotation concentration method, using a mini FLOTAC^®^, for *Toxocara* spp. eggs.

### 2.5. Statistical Analysis

The statistical analysis was performed using SPSS Statistics 24. Means and percentages were calculated and significant correlation (*p* < 0.05) was tested using the independent sample *t*-test.

## 3. Results

The answers from the questionnaires are shown in Table 1. A total of 50 animals (28 dogs and 22 cats) were included in the study. Twenty dogs (71%) and 14 cats (64%) were allowed in the bedroom, 15 dogs (54%) slept on the bed with the owner and 7 dogs (25%) in the bed. For cats, these numbers were 14 (64%) and 8 (36%), respectively. The reasons owners gave for allowing pets in their bedroom or bed were mostly cosiness, being unable to keep the pet out or out of habit. The reasons given by owners who did not allow pets in their bedroom or bed were mainly down to hygiene. Hygiene was mentioned by 42% of all owners, including the owners who did allow their pets in their bed or bedroom.

Most owners were not familiar with the ESCCAP Europe advised frequency of deworming [15]. Many owners did not remember when the last anthelmintic treatment was given or whether they performed flea treatment according to the recommended frequency of the product used.

Out of the dog owners, 12/28 (43%) dewormed correctly with a frequency of 4 times or more per year. This was 4/22 (18%) among the cat owners. In total, 23/28 (82%) of owners dewormed their dogs (average frequency 3.2 times/year) and 21/28 (75%) performed flea control (average frequency 4.3 times/year).

Eighteen cat owners (82%) dewormed their cats with an average frequency of 2.8 times/year and 14/22 (64%) performed flea control with an average frequency of 3.6 times/year.

The aerobic colonies per cm^2^ of sampled fur were calculated by dividing the number of colonies by the estimated surface sampled (1600 cm^2^ for cats, 2500 cm^2^ for dogs) and are shown in Figure 1. The mean aerobic colony count (ACC) is 22.5 cfu/cm^2^ for cats (range 0.5 to 90) and 34.3 cfu/cm^2^ for dogs (range 0.3 to 216). 

The fur of 1 dog and 1 cat tested positive for Enterobacteriaceae, as did the footpads of 18 dogs (64%) and 6 cats (27%) (Table 2). Multiple Enterobacteriaceae colonies (1 to 7) were found on the footpads. Clostridium difficile, *Salmonella* spp. or Campylobacter jejuni were not found in the fur samples. The samples plated on the Campylobacter and blood agar plates did not show any pathogenic bacteria.

*Microsporum canis* was not found on the fur of the animals.

In 7/50 animals (14%), fleas or flea eggs were found in the animal’s sleeping place. These consisted of 2/28 (7%) dogs and 5/22 (23%) cats. *Cheyletiella* spp. mites and *Toxocara* spp. roundworm eggs were not found.

In total, 45% of the cats and 75% of the dogs were positive for one or more potential zoonotic pathogen. No significant correlation was found between the presence of pathogens and owner, animal characteristics, diet, location, parasite control regime or animal washing habits. Flea prevention appeared to negatively correlate with flea absence (*p* = 0.012).

## 4. Discussion

The results of our study show that many owners allow their pets into their bedrooms (68%) and even in their beds (30%). There is nearly no difference between the percentages of dogs and cats that are allowed on the bedroom floor, in the bedroom, on the bed or under the blankets. The reasons to allow dogs and cats in the bedroom or in bed were different between these species. It is very likely that the real number of pets accessing their owners’ bedrooms and beds is even higher, since a lot of pets might do this without the permission or knowledge of their owners. These are higher numbers than found in an earlier Dutch study, where 53% of the animals could sleep on the bed and 24% in the bed [4], suggesting that the number of pets allowed in the bed(room) is increasing. Chomel and Sun reported in 2011 similar results from the USA, France and the UK, where a relatively large number of pets (14–33% of dogs and 45–60% of cats) slept in bed with their owner [8]. This was also found in countries such as the Czech Republic (45%), Qatar (63.3%) and Canada (26% of pets slept with children) [16,17,18].

The increasing number of pets that are allowed in the bedroom is even more surprising since 42% of all the owners mentioned hygiene as a reason not to allow their pets in bed. Therefore, although owners considered their pets a hygiene threat, many owners did not act upon this. These results show a lack of knowledge about the health risks associated with poor hygiene among many owners. This increase also equates to an increased zoonotic risk. That this risk is actually present was reported in a literature review of zoonoses that were obtained by owners after sleeping with healthy pets. Several zoonotic infections, even life-threatening infections, were documented, such as cat-scratch disease, plague, Chagas disease, pasteurellosis, pet bites, *Capnocytophaga canimorsus* septicemia, MRSA infection and cheyletiellosis. Carriage of ectoparasites and endoparasites was considered as a major concern [8].

Most owners in our study did not know how to properly protect themselves and their pets from common zoonotic parasites. The question to owners about the frequency of deworming and flea prevention of their pets appeared to be a difficult question. Moreover, 30% of the dog and cat owners did not use any flea prevention. The negative correlation between the absence of fleas and the use of flea prevention might be caused by incorrect use of the product.

Around 18% of both dogs and cats were fed raw meat in the study. This corresponds with the results of a survey in five European countries among 5000 dogs and cats, where 19% of the dogs and 16% of the cats were fed raw meat [19]. When pets are fed raw meat, this can be contaminated with zoonotic pathogens such as *Salmonella* spp., *E. coli* O157:H7, and extended-spectrum beta-lactamases (ESBL) producing *E. coli* [20,21]. Clostridium perfringens, *C. difficile*, *Staphylococcus aureus* and *Listeria* spp. have been identified in commercial raw pet diets [21,22], resulting in infection risks and antimicrobial resistance of the animal and the owner. When pets are infected with *Salmonella* spp., they may shed *Salmonella* spp. in their faeces and contaminate their fur and the environment such as furniture, carpets or the bed of the owner [23]. Most pets do not show clinical signs and are, as carriers, a hidden source of contamination [24]. Another potential zoonotic risk of a raw meat diet is that it can occasionally be infected with the parasite *Toxoplasma gondii*, however, there is no association of infection in humans after direct contact of infected cats [25].

The impact of finding high numbers of aerobic bacteria on the fur of the animals reflects the level of hygiene, and can be assessed by comparison with microbial guidelines for hospitals or kitchens. These standards are based on the fact that elevated mean ACC (aerobic colony count) lead to higher chances for the presence of pathogenic bacteria. Therefore, on surfaces where food is being prepared and the surfaces of a hospital that are frequently touched by hands (e.g., door handles, light switches and bed linen), the ACC should not exceed 5 cfu/cm^2^ [2,26,27]. The mean ACCs found in our study of 22.5 cfu/cm^2^ for cats and 34.3 cfu/cm^2^ for dogs exceed these maxima 4 to 7 times (Figure 1). From the dogs 23/28 (82%) exceed this level and 15/22 (68%) of the cats. The highest ACCs were 90 cfu/cm^2^ (18x the maximum) for cats and 216 cfu/cm^2^ (43× the maximum) for dogs. These numbers show the importance of proper hygiene when in direct contact with pets.

The presence of pathogenic bacteria is even more important. Enterobacteriaceae (faecal bacteria) that were found in our study can easily be transmitted to the owners via direct (e.g., by stroking, in bed) or indirect contact (e.g., walking over the sheets) [14]. This can be *E. coli* bacteria producing extended-spectrum β-lactamases (ESBL) or AmpC β-lactamase [28,29,30] with the risk of multi-drug resistance [31]. The highly pathogenic serotype *E. coli* O157:H7 can be transmitted by asymptomatic dogs to humans, where it may cause bloody diarrhoeal syndrome [32,33].

*Clostridium difficile* was not found but is considered a zoonotic pathogen [34,35]. Direct contact might be a possible transmission route [36]. In a Dutch study, faeces from 25% of diarrhoeic dogs and 18% of diarrhoeic cats tested positive for *C. difficile* [37], showing a high prevalence of this pathogen.

*Salmonella* spp. can be found in raw meat, but also in soil where it can survive for a year [38]. This is another possible infection route to the animal fur and footpads when they have outdoor access.

*Campylobacter* spp. are commonly isolated from faecal samples of dogs and cats [39]. and transmitted to humans, especially by puppies [40]. While we did not find any evidence of *Campylobacter* spp. present on the fur of the sampled animals, human pathogenic *Campylobacter* rarely causes disease in the pets carrying them [41].

*Microsporum canis* was not found, though it is the most important and most often identified dermatophyte in dogs and cats [42,43]. A study discovered that 17% of asymptomatic dogs and cats were positive for *M. canis* and showed that even asymptomatic dogs and cats might be a major source of dermatophytosis (ringworm) in (mainly) children [44]. Striking increases in ringworm are reported in publications and correctly associated with the presence of pets. However, it has never been suggested that close contact with infected pets in bed may increase the infection risk [7].

Fleas were found in 14% of the animal sleeping places in our study. This number is particularly striking since our study was performed during a period (February and March) when the flea season had not yet started. Presumably, these numbers would be much higher during summer/autumn. A study in the UK in the months April and June revealed that 28% of the dogs and 14% of the cats were flea-infested [45]. With a PCR it was found that 14% of the fleas were positive for at least one pathogen. Most found were *Bartonella* spp. (11%) in samples mainly from cats. The zoonotic tapeworm *Dipylidium caninum* was found in 3% of the fleas, again mainly originating from cats [45]. In a Dutch study, 50% of the dogs and 52% of the cats with fleas were carrying zoonotic pathogens [46]. Important pathogens that can be transmitted by fleas are Bartonella henselae (cat-scratch disease), *Rickettsia typhi*, *R. felis* (flea-borne rickettsioses) and *Yersinia pestis* (bubonic plague) [47,48]. Flea infestation in our study was detected from sleeping places. These are the most densely inhabited places by fleas because they produce eggs mainly during the night [49]. The infection risk for the owner is increased when the pet then sleeps in the bed. A similar risk is realised by Cheyletiella mite infestation, which is the most common fur mite in dogs and cats. It causes pruritic papular lesions on the torso and arms in humans [43,50].

In this preliminary study the sample size is low, and the results may have led to bias. It would be more valuable if the presence of pathogens could be associated with the owner’s health status and medical history. Intestinal parasites were not found, because we did not perform faecal examinations. Parasites such as *Giardia duodenalis*, *Toxocara canis* and *T. cati*, however, are often detected in faecal samples of household dogs and cats and may cause a zoonotic risk [51]. Neither did we discuss the risks of animal bites and bite wound infections. Altogether, our study can be considered a pilot, but initial results show that sleeping with pets may increase the risk of zoonotic pathogen transmission. The presence of several pathogens (hazards) was demonstrated, and exposure will only increase with long-lasting intimate contact when pets sleep in bed with their owners.

## 5. Conclusions

The results of this preliminary study confirm suggestions from literature that pets may pose a potential risk in the transmission of zoonotic pathogens to their owner. This is the case for all pet owners, but risks increase especially for those who have extended exposure and intimate contact with their pets, like sharing a bed, since this highly elevates the exposure rate to pathogens that may be present. Therefore, it is of great importance that owners have a good understanding of how to interact with their pets in a responsible way. This is especially advised for young, old, pregnant or immunocompromised owners, who are at a greater risk if they develop infections. Owners should be informed about these zoonotic risks and educated to interact with their pets in a more responsible way.

## Figures and Tables

**Figure 1 pathogens-11-01149-f001:**
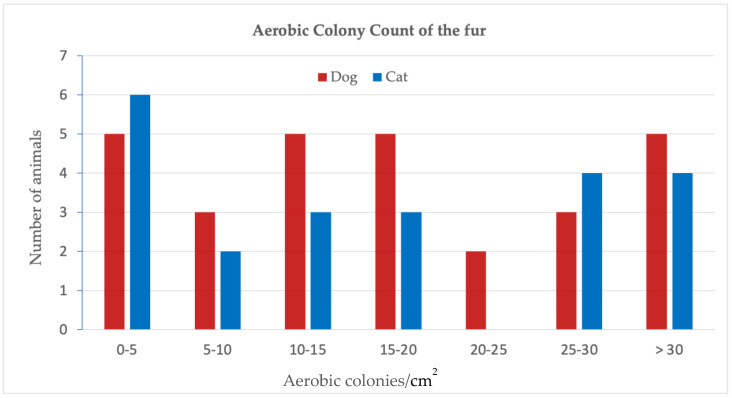
Aerobic colony count of the fur in dogs and cats (should not exceed 5 cfu/cm^2^ in hospitals and on surfaces for food preparation).

**Table 1 pathogens-11-01149-t001:** Questions and answers of the questionnaires *.

	Dog	Cat	Dog + Cat %
Data owner			
-gender	75% F, 25% M	59% F, 41% M	
-age (years)	20–75 (mean 42)	10–68 (mean 41)	
Data pet			
-number	28	22	
-gender	39% F, 61% M	64% F, 36% M	
-age (years)	1–12 (mean 4.8)	1–17 (mean 6.7)	
Sleeping place pet			
-bedroom floor	23	17	80%
-in the bedroom	20	14	68%
-on the bed	15	14	38%
-under the blankets	7	8	30%
Reasons to allow pets in the bed or bedroom			
-out of habit	2	3	10%
-unable to keep out	2	5	14%
-cosiness	7	5	24%
Reasons for not allowing pets in the bed or bedroom			
-hygiene	12	9	42%
House type			
-student dorm room	5	3	16%
-apartment	6	3	18%
-family house	17	16	68%
Outdoor policy			
-outdoors	28	19	78%
-with leash	7 (25%)	N.A.	
-without leash	21 (75%)	N.A.	
Disease clinical signs	1 (eye ulcer)	0	2%
Presence of (zoonotic) disease	1	2	6%
Clinical signs of the owner	(dermatophytosis)	(respiratory, itch)	
Feeding raw meat diet	5	4	18%
Catching prey animals	1 (2%)	9 (41%)	
-once a week or	0 (0%)	2 (9%)	
more often			
Deworming			
-number of owners	23	18	82%
-average frequency	3.2 (1–9)	2.8 (0.5–12)	
per year			
Flea control			
-number of owners	21	14	70%
-average frequency	4.3 (1–13)	3.6 (0.5–12)	
per year			
Washing			
-number of owners	23 (82%)	1 (5%)	
average frequency	11x (1–52)	3x	
-per year			

* Results that were very similar are combined for dogs and cats. N.A.: Not Applicable

**Table 2 pathogens-11-01149-t002:** Animals tested positive on pathogens.

	Dogs	Cats	Total
Enterobacteriaceae on fur	1	1	2
Enterobacteriaceae on footpads	18	6	24
Total Enterobacteriaceae	19	7	26
Fleas	2	5	7

## Data Availability

Not applicable.

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
