# Peer review of "Zoonotic Risks of Sleeping with Pets"

_pathogens, 2022, doi:10.3390/pathogens11101149_

Round 1
Reviewer 1 Report
The manuscript „ The zoonotic risks of sleeping with pets" by Zanen et al. is interesting study from Netherlands showing data on the distribution of cats and dogs pathogens in contact with humans. However, I have a few questions for the authors of this manuscript.
Major revision:
1. I think the title of the manuscript is too extensive and I would suggest changing the title.
2. Why the authors did not check what species of fleas were collected from the tested animals? How many fleas were collected?
3. Are only fleas dangerous to humans for ectoparasites in dogs and cats? Were the animals checked for the presence of other ectoparasites?
Minor review:
Line 30: Are there any other reports of dogs and cats from countries other than Dutch countries?
Line 61: Information should be added which species of fleas may be present in dogs and cats. Only one species of Bartonella is dangerous for human?
Author Response
Point-to-point response to the comments of Reviewer 1.
- I think the title of the manuscript is too extensive and I would suggest changing the title.
- The title has been changed into more general: Zoonotic risks of sleeping with pets.
- Why the authors did not check what species of fleas were collected from the tested animals? How many fleas were collected?
- The assessment of the eventual presence of fleas was the main objective of the study, because in that case there can be considered a zoonotic risk. That depends in our opinion not on the found flea species nor the number of fleas.
- Are only fleas dangerous to humans for ectoparasites in dogs and cats? Were the animals checked for the presence of other ectoparasites?
- No, also other ectoparasites such as Cheyletiella fur mites may be a risk of zoonotic infection. Therefore, we also investigated the debris for this ectoparasite. This is mentioned in the first line under 2.3. Sampling of the animals.
- Line 30: Are there any other reports of dogs and cats from countries other than Dutch countries?
- Yes, this information has been reported in the Discussion (line 221-225).
- Line 61: Information should be added which species of fleas may be present in dogs and cats. Only one species of Bartonella is dangerous for human?
- Information regarding the flea species of veterinary importance and other zoonotic Bartonella spp. has been added to the text
We would like to express our great appreciation to you for the comments on our paper.
Reviewer 2 Report
This article is indeed a great beginning and important database for pet owners and their pets in multiple ways. Authors in this preliminary study shows that pets may constitute a putative risk in the transmission of zoonotic pathogens to their owner. Authors stated that direct contact while sleeping in the same bed possible enhance the risks and directs need to educated owners about zoonotic risks.
Minor corrections
Typo Line-110, regular.
Conclusion Line 276-277 can be modified
Author Response
Point-to-point response to the comments of Reviewer 2.
- Typo Line-110, regular.
- This has been modified
- Conclusion Line 276-277 can be modified
- This has been modified
We would like to express our great appreciation to you for the comments on our paper.
Reviewer 3 Report
This is an interesting study; however, the sample size is low, the study would be more valuable if authors could associate the presence of pathogens with owner’s health status and medical history.
Author Response
Point-to-point response to the comments of Reviewer 3.
- This is an interesting study; however, the sample size is low, the study would be more valuable if authors could associate the presence of pathogens with owner’s health status and medical history.
- The fact that this is a preliminary study with its limitations has been more emphasized in the Discussion and Conclusion